# Micellar Choline-Acetyltransferase Complexes Exhibit Ultra-Boosted Catalytic Rate for Acetylcholine Synthesis—Mechanistic Insights for Development of Acetylcholine-Enhancing Micellar Nanotherapeutics

**DOI:** 10.3390/ijms252413602

**Published:** 2024-12-19

**Authors:** Davide Dante, Jatin Jangra, Anurag T. K. Baidya, Rajnish Kumar, Taher Darreh-Shori

**Affiliations:** 1Division of Clinical Geriatrics, Centre for Alzheimer Research, Department of Neurobiology, Care Sciences and Society, Karolinska Institutet, 141 57 Stockholm, Sweden; davide.dante@ki.se; 2Department of Pharmaceutical Engineering & Technology, Indian Institute of Technology (B.H.U.), Varanasi 221005, India; jatin.rs.phe24@iitbhu.ac.in (J.J.); anuragtkbaidya.rs.phe20@itbhu.ac.in (A.T.K.B.); rajnish.phe@iitbhu.ac.in (R.K.)

**Keywords:** cholinergic signaling, choline-acetyltransferase (ChAT), acetylcholine (ACh), catalytic rate, acetylcholinesterase (AChE), non-ionic surfactants/detergents, Triton X-100, Tween 20, critical micelle concentration (CMC), ChAT–micelle nanoparticles

## Abstract

Choline-acetyltransferase (ChAT) is the key cholinergic enzyme responsible for the biosynthesis of acetylcholine (ACh), a crucial signaling molecule with both canonical neurotransmitter function and auto- and paracrine signaling activity in non-neuronal cells, such as lymphocytes and astroglia. Cholinergic dysfunction is linked to both neurodegenerative and inflammatory diseases. In this study, we investigated a serendipitous observation, namely that the catalytic rate of human recombinant ChAT (rhChAT) protein greatly differed in buffered solution in the presence and absence of Triton X-100 (TX100). At a single concentration of 0.05% (*v*/*v*), TX100 boosted the specific activity of rhChAT by 4-fold. Dose–response analysis within a TX100 concentration range of 0.8% to 0.008% (accounting for 13.7 mM to 0.013 mM) resulted in an S-shaped response curve, indicative of an over 10-fold boost in the catalytic rate of rhChAT. This dramatic boost was unlikely due to a mere structural stabilization since it remained even after the addition of 1.0 mg/mL gelatin to the ChAT solution as a protein stabilizer. Furthermore, we found that the catalytic function of the ACh-degrading enzyme, AChE, was unaffected by TX100, underscoring the specificity of the effect for ChAT. Examination of the dose–response curve in relation to the critical micelle concentration (CMC) of TX100 revealed that a boost in ChAT activity occurred when the TX100 concentration passed its CMC, indicating that formation of micelle–ChAT complexes was crucial. We challenged this hypothesis by repeating the experiment on Tween 20 (TW20), another non-ionic surfactant with ~3-fold lower CMC compared to TX100 (0.06 vs. 0.2 mM). The analysis confirmed that micelle formation is crucial for ultra-boosting the activity of ChAT. In silico molecular dynamic simulation supported the notion of ChAT–micelle complex formation. We hypothesize that TX100 or TW20 micelles, by mimicking cell–membrane microenvironments, facilitate ChAT in accessing its full catalytic potential by fine-tuning its structural stabilization and/or enhancing its substrate accessibility. These insights are expected to facilitate research toward the development of new cholinergic-enhancing therapeutics through the formulation of micelle-embedded ChAT nanoparticles.

## 1. Introduction

The enzyme choline-acetyltransferase (ChAT) (EC 2.3.1.6) is the main acetylcholine (ACh) biosynthesizing enzyme. It defines neuronal and non-neuronal cholinergic cells, i.e., cells that synthesize ACh and use it as an auto- and/or paracrine signaling molecule [1,2,3,4]. ChAT is therefore widely distributed across various tissues, including non-neuronal systems in diverse cell types, such as immune cells, astroglia cells, and sperm, though its canonical functional significance lies within the central nervous system (CNS) [1,2,3,5]. In the CNS, ChAT activity is crucial for maintaining cholinergic neurotransmission, influencing processes such as memory consolidation, attention, and other cognitive functions. Acetylcholinesterase (AChE) is similarly expressed in a wide range of tissues, including neurons, red blood cells, and lymphocytes [2,4,6,7]. These enzymes are crucial for cholinergic neurotransmission, with AChE breaking down acetylcholine in synaptic clefts and ChAT catalyzing the synthesis of acetylcholine [8]. Proper functioning of these enzymes is essential for maintaining cholinergic balance, and their dysregulation is linked to neurodegenerative diseases, including major dementia disorders, such as Alzheimer’s disease (AD), in which the major therapeutic options are still cholinesterase inhibitors (ChEIs). These drugs are used to counter the cholinergic deficit in AD by preventing the degradation of ACh by AChE [9]. Nonetheless, ChEIs show limited clinical efficacy perhaps because ACh biosynthesis is not optimal in the cholinergic neurons, which calls for a new strategy, where the catalytic function of ChAT is boosted to counteract the observed reduction in the neuronal expression of ChAT [10,11]. Intriguingly, beta-amyloid peptide 42 (Aβ_42_) is the first documented ChAT activity booster, which are termed ChAT potentiating ligands (CPLs) [10,11]. In addition, ChAT has been targeted for development of specific cholinergic biomarkers, for instance, for the development of positron emission tomography (PET) imaging probes to map early cholinergic changes in various cholinergic-related neurodegenerative diseases, including neuromotor disorders such as amyotrophic lateral sclerosis (ALS) and dementias such as AD [12,13]. All of these require early identification to optimize the chances of finding a viable therapeutic strategy [12,13]. As PET imaging probes, ChAT ligands have significant potential for the early identification of AD and related dementias, including Down syndrome and Lewy body diseases [12,14].

The distribution of ChAT has been measured within the neuron, and it appears to be 80–90% as cytosolic protein, a portion of which can be ionically bound to the membrane, and 10–20% of the enzyme appears to be non-ionically bound to the plasma membrane [5]. A recent report indicates that ChAT may also be in the sperm membrane extracellularly [3]. In our work with ChAT, we have encountered some unexplained in vitro changes in the activity of recombinant human ChAT protein in various buffer systems containing different detergents or different concentrations of the included detergents such as Triton X-100 (TX100) and Tween 20 (TW20).

Detergents are indispensable components of many buffer systems used in biochemical and biomedical research due to their remarkable ability to solubilize membrane proteins while preserving their activity in solution [15,16,17]. The use of detergents spans a wide range of applications, from facilitating the extraction of membrane-bound proteins, such as certain enzymes or receptors, to maintaining the structural stability and functions of the proteins for various purposes [16,18,19]. Two of the most common detergents in life science research are the non-ionic detergents TX100 and TW20 [20,21,22,23]. This is largely due to their mild nature, which minimizes denaturation and preserves the native structure and function of proteins. Nonetheless, some studies indicate that TX100 can alter the function of certain enzymes [22,24,25,26]. Some studies indicate that the functional alteration of the enzymes may be related to the micellar properties of the detergent [22,24,25,27]. Others suggest that the changes in the activity may be caused by an increase in the solubility of the enzymes [28,29].

Triton X-100 has on average a molecular weight of ~625 g/mol and a critical micelle concentration (CMC) of 0.2–0.3 mM, with an aggregation number typically ranging between 75 and 165 molecules. Commonly, TX100 is added to buffers at 0.05% (*v*/*v*) concentration, while its concentrations range from 0.1% to 0.6% in buffers for solubilizing membrane proteins while preserving the structure. Tween 20 has a higher molecular weight of ~1228 g/mol than TX100, but its CMC (0.06–0.07 mM) is ~4 times lower than the CMC of TX100, with an aggregation number of 60 [23,30].

In this study, we investigated the in vitro changes in ChAT activity observed in the most common buffers with and without TX100 and TW20. We show here that increases in ChAT catalytic efficiency occur mainly at post-CMC concentration of the detergent regardless of the buffer system. We also show that the boosted ChAT catalytic function is unlikely to be related to an improved solubility of ChAT protein or stability of its tertiary structure. Rather, the results suggest that membrane-like microenvironment formation accounts for the altered ChAT activity because ChAT becomes catalytically ultrafast depending on the post-CMC concentrations of TX100 or TW20. This report delivers crucial insights into the function of an enzyme that is implicated in neurodegenerative diseases, as well as opens a new window into a cholinergic-enhancing strategy by introducing the delivery of ultrafast ChAT via functionally enhanced nanoparticles [31].

## 2. Results

### 2.1. TX100 and TW20 Differentially Affect the Catalytic Rate of ChAT in a TBS Buffer System

We first tested the ChAT activity in the TBS buffer, as it is the most commonly used buffer in our lab. We then tested the effect of two common non-ionic detergents, namely TX100 and TW20, on ChAT activity in the TBS buffer at a single concentration of 0.05% (*v*/*v*) for both detergents, alone and in combination. As control, we used TBS without the detergent. ChAT activity in terms of conversion of acetyl-CoA and choline to -CoA and acetylcholine was then monitored continuously for 30 min. The data are presented in Figure 1A. As can be appreciated from Figure 1A, in the absence of TX100 or TW20 in the TBS buffer, the ChAT activity seems to follow two dynamic phases with a high initial rate for about six minutes (the initial phase, *I-Phase*, Figure 1B), followed by an abrupt reduction in the catalytic rate of ChAT (denoted here as the late phase or *L-Phase*). In contrast, in the presence of 0.05% TX100 in the TBS buffer, the ChAT catalytic rate was constant during the whole 30 min monitoring time. Nonetheless, in the presence of TW20, the catalytic rate of ChAT was not constant as it was in the presence of TX100, but followed a two-phase dynamic, similar to that observed in TBS without detergent. However, the effect of 0.05% TW20 in the buffer was in terms of prolonging the catalytic rate of ChAT in both the *I-Phase* (from 6 to 9–12 min) and *L-Phase* (Figure 1C).

It should be noted that the observed reduction in the catalytic rate of ChAT in the L-phase is unlikely to be due to exhaustion of ChAT substrates (choline and acetyl-CoA), since ChAT activity remains unchanged for instance for TBS+TX100 (Figure 1A). We also investigated how a combination of TX100 and TW20 in the TBS buffer affect the activity of ChAT. The results indicated that in this condition, ChAT activity remained unchanged through the whole 30 min monitoring time, like the TBS with TX100 alone. Nonetheless, the ChAT rate was greatly reduced when TX100 and TW20 were present together in the TBS buffer (Figure 1A–C). Overall, the results indicated that TX100 and TW20 in TBS caused differential dynamic changes in ChAT activity over time in terms of significant variation in the catalytic function of ChAT.

### 2.2. The Effect of TX100 and TW20 on the Catalytic Rate of ChAT in HEPES Is Largely Like in the TBS Buffer System

We then repeated the experiment in another common buffer, namely HEPES buffer, to investigate whether the effect of TX100 and TW20 was buffer system-dependent or not. The results indicated that the effect of TX100 in the HEPES buffer on the catalytic rate of ChAT was identical to that in the TBS buffer (compare the purple graphs in Figure 1D with Figure 1A). In other words, the catalytic rate of ChAT remained completely unchanged in the HEPES buffer with 0.05% TX100. Similarly, the effect of the combination of TX100 and TW20 on the catalytic rate of ChAT was essentially identical to that in TBS (compare the black graphs in Figure 1D–F with Figure 1A–C). Nonetheless, there were two differences. Firstly, TW20 had much less boosting effect on the catalytic rate of ChAT in the HEPES than in the TBS buffer (compare the orange graphs in Figure 1D–F with Figure 1A–C). Secondly, ChAT seemed to be inactive in the HEPES buffer lacking TX100 or TW20 (compare HEPES Ctrl with TBS Ctrl graphs in Figure 1D with Figure 1A).

### 2.3. Quantitative Analysis Confirms the Boosting Effect of TX100 and TW20 on the Catalytic Rate of ChAT in HEPES and TBS Buffer Systems

The quantified changes in the catalytic rate of ChAT in the two buffer systems with and without the detergents TX100 and TW20, as well as their combinations, are presented in Figure 2. The results clearly show that ChAT activity is greatly boosted by 0.05% of TX100 in both HEPES (~4.5-fold) and TBS (~4-fold). The results also confirm that the effect of TX100 was independent of the buffer system. The boosting effect of 0.05% TW20 was less than the effect of TX100 in both buffer systems (~2-fold in HEPES and ~2.5-fold in TBS; Figure 2). In addition, the boosting effect of TW20 on ChAT activity was short-lived in the HEPES buffer given that it was greatly reduced during the *L-phase* compared to the *I-phase* in the 30 min monitoring window. The result also confirmed that although the combination of TX100 and TW20 in the buffer still greatly boosted the ChAT activity compared to the control conditions (i.e., no detergent in the buffers), the boosted effect was much less than TX100 alone (5–6-fold) compared to TX100 + TW20 (~3-fold; Figure 2).

### 2.4. The Boosting Effect of TX100 on the Catalytic Rate of ChAT Is Concentration-Dependent

Next, we wanted to investigate whether the boosting effect of the detergents on ChAT activity was concentration-dependent. We prepared a serial dilution solution of TX100 with a concentration range of 3.2 mM to 0.0031 mM (or 0.8% to 0.0008%) in both HEPES and TBS and assessed the changes in the catalytic function of ChAT. The results are summarized in Figure 3A. A simple calculation of the percent boosted ChAT level indicated 57-fold and 129-fold changes in ChAT specific activity in the HEPES and TBS buffer, respectively. The results demonstrate the dramatic boosting impact of TX100 concentrations on ChAT enzymatic activity in a clear dose–response manner, which reaches a plateau at the highest concentration of TX100, i.e., around 0.8% (*v*/*v*).

### 2.5. The Effect of TX100 on the Catalytic Rate of ChAT Is Not Due to Structural Stabilization

Another question that we wanted to investigate was whether the observed boost in the activity of ChAT was due to a structural stabilization of ChAT protein, given that we are using highly purified recombinant ChAT protein. Furthermore, we wanted to observe whether the effect of TX100 was mediated by preventing adsorption of the purified ChAT protein to the wall of the wells of the microtiter plate. To assess these questions, we added a high concentration of gelatin (1.0 mg/mL) to the buffer systems as both a protein stabilizer and a blocking agent for unspecific binding sites [10]. The results are shown in Figure 3B. The results indicate that, to some degree, the addition of gelatin in the buffers increased the basal activity levels of the ChAT protein. This is appreciated by comparing the ChAT activity in the control’s graphs (i.e., *HEPES,* or *TBS Ctrl*) in Figure 3A with the control values in Figure 3B (i.e., *HEPES + Gelatin Ctrl and TBS + Gelatin Ctrl*).

Nonetheless, this did not account for the ultra-boosted activity of the ChAT protein by TX100, as is clearly appreciated from the characteristic S-shaped curve with both HEPES/TBS–gelatin buffers. Thus, the presence of gelatin—a known protein stabilizer—did not significantly alter the detergent-driven activation pattern (compare the S-shaped graphs in Figure 3A,B), underscoring the dramatic boosting effect of TX100 on the specific activity of ChAT. As before, a gross calculation of the overall changes in the basal ChAT activity level indicated that the ChAT catalytic rate was boosted by 8–10-fold over the basal detergent-free ChAT activity levels when the buffers contained gelatin. Thus, we concluded that the boosted ChAT activity by TX100 was unlikely to be simply due to a stabilization of the tertiary structure of the ChAT protein by TX100, nor was it mediated by reducing the adsorption of ChAT protein to the surface of the wells of the microtiter plate.

### 2.6. The Boosting Effect of TX100 on the Catalytic Rate of Enzyme Is Specific to ChAT Protein

Next, we asked whether the boosting effect of TX100 is specific to ChAT protein. Therefore, we performed the same experiment on recombinant human AChE protein, an enzyme that is closely linked to ChAT, in terms of regulating the levels of ACh. The results of this experiment are presented in Figure 3C. The results show no significant pattern of changes in the AChE activity, except some apparently random activity fluctuations. Therefore, we may conclude that the experiment indicated that the activity-boosting effect of TX100 is highly specific to the ChAT enzyme compared with AChE.

### 2.7. The Boosting Effect of TX100 and TW20 on the Catalytic Rate of ChAT Is Associated with the Micellization Properties of These Two Detergents

The observed enzyme-specific S-shaped behavior of ChAT activity in response to concentrations of TX100 seems to emphasize a certain property of detergents, such as TX100, namely the critical micelle concentration (CMC). The CMC for TX100 is 0.2 mM. An examination of the S-shaped dose–response curves of TX100 for ChAT activity with respect to the CMC of TX100 demonstrates almost no changes in ChAT activity until TX100 concentration passes its CMC value (as depicted in Figure 4A), regardless of the buffer system. Therefore, we hypothesized that if this is true, then the same should occur with regard to CMC of TW20, another non-ionic surfactant, particularly given that the CMC of TW20 is 0.06 mM, which is about 4-fold lower than CMC of TX100. We investigated this hypothesis by assessing a dose–response experiment on a serial concentration solution series of TW20 around its CMC, namely 3.8 mM to 0.0038 mM. The results are shown in Figure 4B, in which the CMC of TW20 is specified. The result is identical to those of TX100, showing that the activity of ChAT indeed increases when the concentration of TW20 passes its CMC value. Figure 4C provides a schematic illustration of our hypothesis, which is that TX100 and TW20 produce micelles that act as a membrane-like carrier structure/platform for ChAT protein. Apparently, a consequence of incorporation of ChAT protein in such a membrane-like microenvironment is that ChAT protein gains an ultrafast catalytic property, through a yet unknown mechanism.

### 2.8. In Silico Molecular Simulation Suggest ChAT Show High Embedding Affinity for Micelles

To provide support for our hypothesis, we assessed the structural stability and dynamic properties of the ChAT protein in a construct of bilayered TX100 micellar membrane complex by means of molecular dynamic (MD) simulations conducted over a 5 ns period. The spatial configuration of the overall protein–membrane complex along the MD simulation is depicted in sequential snapshots in Figure 5A, illustrating a close association of ChAT with the micelle over time. Nonetheless, the simulation results indicated stable behavior of the ChAT–micelle complex, as deduced by the conventional MD metrics, namely RMSD, RMSF, RoG, and SASA, as these remained consistent throughout the simulation (Figure 5B–E). The root mean square deviation (RMSD) of the ChAT protein averaged 0.19 nm, demonstrating minimal structural deviations and indicating a stable interaction at the micelle interface (Figure 5B). Similarly, TX100, being the component of the membrane exhibited, despite minor adjustments, a relatively stable but higher average RMSD of 2.95 nm (Figure 5B). Root mean square fluctuation (RMSF) analysis of ChAT protein revealed an average value of 0.12 nm, further emphasizing the overall stability of the ChAT protein with limited flexibility, and slight fluctuations concentrated around loop regions that are likely necessary for functional flexibility (Figure 5C). The radius of gyration (RoG) averaged 4.55 nm, indicating that a compact structure was maintained by ChAT across the simulation period (Figure 5D). The stable RoG, combined with a consistent solvent-accessible surface area (SASA) of on average 264.30 nm^2^ (Figure 5E), suggests a limited solvent exposure, which is in agreement with a micellar environment that has effectively encapsulated the protein, supporting its structural integrity.

Overall, the spatial configuration of the overall protein–membrane complex over time, together with the quantitative metrics, supported our hypothesis that micelle formation might provide a favorable microenvironment that mimics the cellular membrane conditions, granting ChAT with structural stabilization, solubilization, potentially enhanced compartmentation, and exposure of catalytic sites to the substrates. In other words, the micelle encapsulation of ChAT results in the formation of facilitated ChAT–micelle nanoparticles.

## 3. Discussion

We show for the first time that ChAT becomes hyper-boosted in the presence of non-ionic detergents such as TX100 and TW20. The boost in the catalytic rate of ChAT by TX100 reached a plateau at ~10-fold over the basal levels in buffer supplemented with 1 mg/mL gelatin as a protein structure stabilizer. We further showed that the observed boost was related to the critical micelle concentration (CMC), since TW20, like TX100, boosted the catalytic rate of ChAT, and in both cases, it happened when their concentrations in the buffers passed their specific CMC, highlighting the enzyme’s specific sensitivity to micelle-induced environmental changes.

To explain these observations, we hypothesized that the boosting effect of TX100 and TW20 on the catalytic rate of ChAT is mediated through the formation of membrane-like micellar structures, which occurs as soon as the concentration of the surfactants exceed their CMCs. Reports have shown that TX-100 micelles follow a multilayer organization, leaving a large part of the hydrophilic tail in the outer region of the micelle, with a strong interaction with surrounding water molecules [23,32,33]. Our results thereby indicate that ChAT becomes embedded into these micellar structures. These provide a microenvironment resembling the native cellular environment, in which ChAT naturally possesses a high specific activity. This is in line with several reports, indicating that ChAT exists as both a soluble cytoplasmic variant and a membrane-bound form. The difference between membrane-bound and membrane-free ChAT forms has been suggested to arise from post-translational modifications [34,35]. For instance, three putative membrane-associated forms of ChAT have been identified across animal models, indicating translocation to the plasma membrane, unlike the cytosolic form. ChAT also seems to exhibit certain peculiarity in a membrane-like environment, for instance, being less subjected to inhibition by inhibitors of the enzyme [36,37]. It is also shown in several animal species that ChAT protein is compartmentalized within synaptosomes, particularly at the presynaptic terminal [37,38]. Therefore, ChAT protein seems to have affinity to membrane-like microenvironments, such as presynaptic membranes, myelin fragments, and synaptosomes, where it exhibits the highest specific activity [38].

We also performed an identical experiment on AChE, another enzyme related to acetylcholine metabolism. We found no changes in the catalytic rate of AChE in the presence of various concentrations of TX100, including CMC, reinforcing the idea that the boosting effect was exclusive to ChAT compared to AChE. Nonetheless, there are reports about other enzymes that their activity is affected by TX100 [20,21,22,24,25,27,39]. For instance, phospholipase-D, another CNS-related membrane-bound enzyme, shows enhanced activity by TX100, which reaches a maximum level at TX100 concentrations of 0.1–0.2% (*w*/*v*) [40]. Notably, when expressed as a percentage, the CMC of TX100 is 0.0125% (or 0.21 mM).

Other enzymes, on the other hand, can be inhibited by TX-100. For instance, cytochrome c oxidase, an enzyme involved in the mitochondria respiratory chain of the Krebs cycle, is inhibited by TX100, with an inhibition constant (*Ki*) of 0.3 mM. Nonetheless, in this case, the effect seems to be mediated by suppression of intraprotein electron transfer by a blocking interaction of TX100 at the catalytic mouth of the enzyme [41]. Similarly, reports indicate that TW20 and its related surfactants readily produce micelles [42,43,44,45]. There are also reports for using these as delivery systems [46] and for preparation of enzyme-loaded nanoparticles (NPs) for enzyme-replacement therapy [47]. For instance, the preformulation of the β-glucosidase (β-Glu) enzyme with TW20, 60, or 80 has been compared for the loading and functionality of poly(lactic-co-glycolic) acid (PLGA) NPs. Intriguingly, the report indicates that the enzyme in the β-Glu-TW20 formulation of the PLGA NPs maintained the longest and exhibited the highest enzyme activity, although all the formulations had the same amount of the enzyme [47]. There is also a report showing that firefly luciferase activity is boosted 7-fold by several micelle-forming surfactants, including TX100 and TW20 [48].

ChAT can gain enhanced activity in other ways as well. For instance, reports show that amyloid-β (Aβ) peptides, in particular Aβ_42_, boosts the activity of human ChAT by ~25% through a direct interaction with ChAT protein [10,11]. This has resulted in the generation of a new class of small molecules, termed ChAT-potentiating ligands (CPLs) [10,11]. The activity of ChAT may also be boosted by phosphorylation at specific residues, such as serine 440 and threonine 456 [35]. In addition, the phosphorylation of ChAT seems to alter its binding to plasma membrane and interaction with other cellular proteins [35]. Notably, all the aforenoted modes of activation of ChAT are modest in their magnitude compared to the 10-fold ultra-boosting of ChAT protein by both TX100 and TW20.

To test our hypothesis, we also performed an in silico molecular dynamics simulation, with the assumption that TX100 can form a membrane-like bilayer construct, which previous reports have indicated [32]. The simulated spatial configuration together with the quantified MD metrics supported our hypothesis that micelle formation provides a favorable microstructure that mimics the cellular membrane conditions, potentially enhancing compartmentation of ChAT. Thereby, the formation of facilitated ChAT–micelle nanoparticles through encapsulation of ChAT by TX100 or TW20 is the most plausible explanation for the results of the current report. This might represent a form of micellar polymer encapsulation of enzymes [33].

Nonetheless, other reports exist that seem to suggest that micelle-induced activity boost may be a common feature among transferases. For instance, a report shows that the mitochondrial carnitine palmitoyl-transferase also gains boosted activity in the presence of TX100 and TW20 concentrations over their CMC values [49]. Another report shows that an ADP-ribosyltransferase from erythrocytes gains several-fold increases in its activity in the presence of TX100 and a zwitterionic detergent [50]. Thus, micellar-induced enzyme activity boost by various surfactants and/or fatty acids warrants further consideration and investigation, particularly since such micellar-embedded enzyme formulation has been shown to be an effective way to generate functional NPs with therapeutic potentials [31,47,51].

In this context, micelle-embedded hyper-boosted ChAT nanoparticles may have therapeutic properties. Reports indicate that acetylcholine, in addition to its canonical function as a neurotransmitter, possesses a strong anti-inflammatory function, which can effectively regulate immune responses through activation of α7 nicotinic receptor [1,52]. A report has also shown that ChAT exists in soluble form in both human plasma and cerebrospinal fluids [2]. Evidence indicates that the function of extracellular ChAT is to maintain an extracellular acetylcholine equilibrium for regulation of the function of immune cells in circulation and astroglial cells in the brain [2]. It has also been reported that the extracellular acetylcholine equilibrium in the brain of patients with AD is dysfunctional due to an interaction between Aβ peptides and cholinesterase, which leads to the formation of complexes, called BAβACs, in which the cholinesterases are hyperactive [2,53]. This results in a shift in the acetylcholine equilibrium, resulting in overactivation of astroglial cells and neuroinflammation [2,53]. It has been shown that cholinesterase inhibitors can partially restore balance, which also correlates with improved cognition in the treated patients [54,55]. We hypothesize that micelle-embedded hyper-boosted ChAT nanoparticles should be able to restore or even elevate the extracellular acetylcholine equilibrium in favor of an anti-inflammatory milieu through an effective suppression of astrogliosis.

This study has several limitations. Firstly, we investigate the effect of only the two most common laboratory detergents. Future investigation with other non-ionic surfactants such as Brij, as well as various ionic and zwitterionic surfactants, may provide insights for deducing the mechanism for the boosting of ChAT catalytic function. Similarly, new studies are warranted to expand the scope of this study to include other natural fatty acids and lipid bilayer constructs as alternatives. Overall, studies are warranted to elucidate whether the observed effects are unique to non-ionic detergents or extend to broader micelle-forming agents. While the in silico molecular dynamics (MD) simulation supported our hypothesis from the in vitro findings, they should be considered preliminary. Therefore, deeper and longer MD simulation than 5 ns is required for investigating the interactions between detergent constructs and ChAT. Additionally, the focus on recombinant human ChAT leaves open the question of whether similar effects would be observed with ChAT from rodent models or from cellular extracts, which may differ in their structural properties and responses. Lastly, while gelatin served effectively as a stabilizer in our experimental setup, it remains unclear whether the result would stand if a naturally occurring carrier protein, such as serum albumin, had been used instead of gelatin. Notably, we could not use bovine serum albumin (BSA) since it is incompatible with our ChAT assay detection systems [10].

Finally, it might be noteworthy to share our experience, which led us to initiate the current investigation. In our work on characterizing new ligands of ChAT, we often encountered dramatic changes in the rate of ChAT, which were difficult to explain. This happened most often when we prepared a new batch of TBS buffer. We tracked this to the really small differences that could occur during the pipetting of TX100 into the buffer, due to the high consistency of the TX100 stock solution. Nonetheless, these discrepancies in ChAT activity mostly affected the characterization of the pharmacodynamic parameters of a new class of ChAT ligands, which we call ChAT-potentiating ligands or CPLs. Given that CPLs are designed to boost the activity of ChAT, the effect of TX100 resulted in substantial variations in determining the potency of the CPLs. Further scrutiny of the phenomenon also indicated that somehow the effect of TX100 is more unpredictable in TBS than in HEPES buffer. Indeed, as can be appreciated from Figure 1, replacing TBS with HEPES provided a much more stable condition for the ChAT assay, which has now been established in our lab.

## 4. Materials and Methods

### 4.1. Chemicals

The following material, all purchased from Merck (Stockholm, Sweden), was used: HEPES (Cat #H700), Trizma base (Cat #T1503), EDTA (Cat #324503), NaCl (Cat # S9888), gelatin bovine skin (Cat #G9391), Triton X-100 or 1,1,3,3-tetramethylbutyl)phenyl-polyethylene glycol [Cat #93443, aggregation number 100-155, average micellar molecular weight (Mw) of 80,000, or an average Mw of 625], Tween 20 or polyethylene glycol sorbitan monolaurate [Cat #P1379; Mw ~ 1228, and CMC of 0.06 mM], choline chloride (Cat #C7527), acetyl-CoA (Cat #A2056), acetylthiocholine iodide (ATC; Cat #A5751), and DTNB [or 5,5′-dithiobis (2-nitrobenzoic acid); Cat #322123].

### 4.2. Production and Purification of Recombinant Human ChAT Protein

Recombinant human ChAT (rhChAT) protein was produced and purified by the Protein Science Facility (PSF) at Karolinska Institute/SciLifeLab (http://ki.se/psf, accessed on 15 December 2024), as described before [56]. Briefly, the purity of the protein was determined using sodium dodecyl sulfate polyacrylamide gel (SDS-PAGE) stained with Coomassie blue dye. The total protein concentration was measured using a DC protein assay (BioRad, Stockholm, Sweden). The storage buffer for the protein was 20 mM HEPES buffer, pH 7.5, containing 300 mM NaCl and 0.5 mM TCEP. The protein was diluted in the storage buffer to a concentration of 212 μg/mL. The diluted enzyme solution was then aliquoted (10 μL/tube), frozen on dry ice, and stored at −20 °C.

### 4.3. Serial Solutions of TX100 and TW20 in the HEPES and TBS Buffer Systems

HEPES buffer (20 mM; pH 7.4, containing 1.0 mM EDTA and 150 mM NaCl). TBS buffer (10 mM; pH 7.4, containing 1.0 mM EDTA and 150 mM NaCl). HEPES– and TBS–gelatin buffers were prepared by adding 1.0 mg/mL (*w*/*v*) of gelatin to these buffers.

A set of TX100 solutions was prepared with 2-fold serial dilution from a 3.2% solution in the HEPES or TBS buffer, ranging from 3.2% to 0.0031%. These solutions were used to achieve 4-fold lower final concentrations in the wells, which ranged from 0.8% to 0.0008%. These TX100 solution series were prepared using an average molecular weight of 625 g/mol, so that a final molar concentration range of 13.7 mM to 0.0134 mM was reached in the wells (final volume of 80 μL).

Similarly, a set of TW20 solutions were prepared with 2-fold serial dilution from a 1.6% (*v*/*v*) solution in the HEPES or TBS buffers, ranging from 1.6% to 0.0016%. These solutions were used to achieve 4-fold lower final concentrations in the wells, which ranged from 0.4% to 0.0004%. Using an average molecular weight of 1227 g/mol, these account for a final molar concentration of TW20 in the wells, which ranged from 3.84 mM to 0.0038 mM in the wells (final volume of 80 μL).

Notably, upon addition of 20 μL of the TX100 or TW20 solution series into the wells of the microtiter plate, the plate was placed in the plate reader and subjected to 30 s orbital shaking at intensity level 5. Following the dispensing of all components, the plate was placed on a shaker and subjected to a second round of orbital shaking for 15 min at 600 rpm to ensure thorough mixing of the added TX100 or TW20 solutions with the choline and the enzyme solutions added in the previous steps. Finally, the plate was placed in the plate reader and 20 μL of cocktail-A was added to each well. The changes in fluorescence intensity were monitored as noted below.

The controls for the HEPES– or TBS–detergent solutions were simply the buffers lacing the detergents (i.e., TSB or HEPES buffers). The enzyme activity in the control wells was used as reference values to compute percentage changes in the ChAT activity by TX100 or TW20. Blanks were wells that contained all components but the enzyme protein (the buffer was used instead). In the single-detergent concentration assessments, the samples were applied in 14 replicates, while in the dose–response assessments, the samples were applied in at least quadruplicate. The controls were, in all cases, over 20 replicates. The final concentration of rhChAT protein was 53 ng/mL, which equals 4.24 ng of rhChAT protein per well.

### 4.4. In Vitro Fluorometric ChAT Activity Assay

In all the analyses, an in-house developed continuous high-throughput fluorometric method was used together with the rhChAT protein [56]. The assay was conducted in 384-well Greiner flat black plates (Greiner Item-No. 781209). The protocol involved the sequential addition of 20 μL/well of 600 μM choline chloride solution (final concentration, Cf, 150 µM), 20 μL/well of the detergent solutions (TX100 and/or TW20) at certain concentrations, and then 20 μL/well of a 0.212 µg/mL of a rhChAT protein solution (Cf = 0.053 μg/mL). The plate was then incubated at room temperature on an orbital shaker at 100 rpm for 30 min.

Finally, 20 µL/well of a cocktail-A [dilution buffer containing 53 μM acetyl-CoA (Cf, 13.3 µM) and 60 μM CPM (Cf, 15 µM)] was added to each well. The changes in fluorescence intensity were monitored for 30 min at 2–3 min intervals using a microplate spectrophotometer reader (Infinite M1000, Tecan Nordic AB, Stockholm, Sweden). The excitation and emission wavelengths were 390 nm and 479 nm, respectively.

### 4.5. In Vitro Colorimetric AChE Activity Assay

An in-house high-throughput assay for the enzymatic activity of AChE was designed using a modified version of Ellman’s colorimetric assay, as described [57]. Purified recombinant human AChE (rhAChE) protein (Cat no. C1682; Merck, Stockholm, Sweden) was used. Briefly, 20 μL/well of a 1:768 diluted solution purified rhAChE protein (Cf = 3.5 ng/mL) was added to the wells of a 384-well plate (a flatbottom transparent plate). Then 20 μL/wells of the serial solutions set of TX100 in HEPES or in TBS was added to the assigned wells (all in quadruplicates), followed by adding 20 μL/well of a 1.6 mM freshly prepared solution of DTNB (Cf = 0.4 mM) to all wells. The plate was incubated at 600 rpm on an orbital shaker at RT for ~15 min. On each 384-well plate, several enzyme wells without TX100 but the buffer vehicle were also included to serve as reference enzyme control wells. Negative controls (or blanks) were wells without the enzyme. Lastly, 20 μL of a 2.0 mM ATC (Cf = 0.5 mM) was added to each well, and the changes in absorbance were continuously monitored at 412 nm wavelength for 10 min with 1 min intervals using the microplate spectrophotometer reader (Infinite M1000, Tecan). The rate of enzyme activity was determined from the linear part of the kinetic reaction curves as ΔOD/time. The total volume in all wells was 80 μL. The percentage changes for each TX100 concentration was calculated based on the enzyme control value as a reference (100% activity).

### 4.6. Molecular Dynamics (MD) Simulation Study on ChAT with TX100

Molecular dynamics (MD) simulations were conducted to investigate the stability and interaction dynamics between ChAT and a bilayered micelle membrane composed of TX100 molecules. Using the GROMACS 2020 software package, the CHARMM36M force field was employed to model atomic interactions within this protein–membrane system [58]. The initial structure of the protein was obtained from the Protein Data Bank (PDB ID: 2FY3) and assessed for missing residues with simultaneous removal of redundant solvents, ions, or molecules. To focus on the interaction behavior between the protein and the membrane, the protein was positioned just above the upper leaflet of the TX100 bilayer micelle using CHARMM-GUI, an approach designed to assess both the stability of the system and the interactions occurring at the protein–membrane interface [59].

The bilayer micelle was generated following CHARMM-GUI protocols, specifically tailored to heterogeneous bilayer systems, and the system size in the XY plane was determined based on the protein’s cross-sectional area along the *Z*-axis, ensuring complete coverage of the micelle around the protein. The TIP3P water model was used for solvation, providing an accurate model for water behavior in a biological environment. A 0.15 M NaCl concentration was introduced to neutralize the system using the Monte Carlo ion placement method, ensuring electrostatic stability. Periodic boundary conditions were applied to simulate an infinite system, and the complete protein–membrane complex was enclosed within a cubic simulation box.

To prepare the system for simulation, an initial energy minimization was carried out using the steepest descent algorithm for 500,000 steps, addressing steric clashes and unfavorable interactions. Hydrogen bonds were constrained using the LINCS algorithm to stabilize bond lengths during both the equilibration and production phases [60]. Two equilibration steps followed the energy minimization, viz. a 1-nanosecond NVT phase, conducted under the canonical ensemble, and a 1-nanosecond NPT phase, conducted under the isothermal-isobaric ensemble. Temperature control was maintained at 310.15 K using the velocity-rescaling method, while the pressure was regulated at 1 bar using the Parrinello–Rahman barostat to replicate physiological conditions.

The production MD run was subsequently performed for 5 nanoseconds under constant temperature and pressure, with the Nose–Hoover thermostat and Parrinello–Rahman barostat employed for temperature and pressure control. Non-bonded interactions were managed using the Verlet force-switch function, with cutoffs set at 1.0 nm and 1.2 nm for Lennard–Jones interactions, while the particle mesh Ewald (PME) method was applied to account for long-range electrostatic interactions. Furthermore, analysis of the simulation trajectories included calculations of root mean square deviation (RMSD) and root mean square fluctuation (RMSF) to evaluate the stability and flexibility of the system. Additionally, structural insights were obtained by analyzing solvent-accessible surface area (SASA) and radius of gyration (RoG) using the GROMACS modules gmx_sasa and gmx_gyrate, respectively. Visualization of trajectories was performed using VMD 1.9.4. software, and clear graphical representations of simulation results were drawn using the QtGrace tool [61,62].

## 5. Conclusions

Our findings reveal that a unique Triton X100 and Tween 20 micelle-induced hyperactivation of ChAT, distinct from the behavior of other enzymes such as AChE, has potentially far-reaching implications for enzyme-based therapeutics and drug delivery strategies. Future studies should aim to clarify the exact mechanisms by which micelle environments modulate this key cholinergic enzyme and explore the potential of micelle-forming compounds as both therapeutic enhancers and delivery agents.

## Figures and Tables

**Figure 1 ijms-25-13602-f001:**
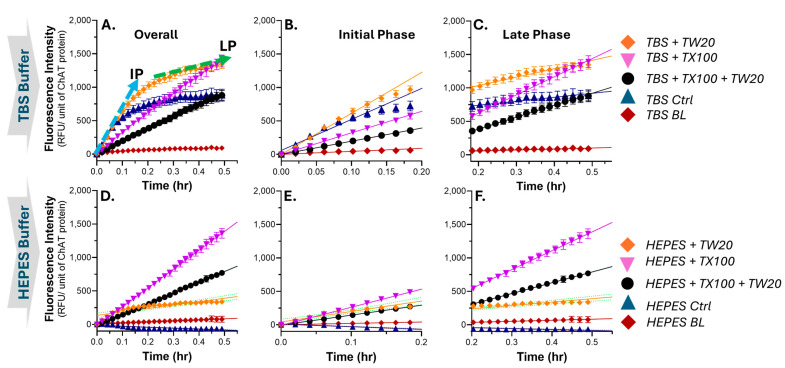
Continuous real-time monitoring of changes in fluorescence intensity over time as a measure of ChAT activity in different buffer-detergent conditions. (**A**–**C**) Average increase in fluorescence intensity (FI) as a function of time as a measure of catalytic rate of ChAT protein in TBS buffer system with and without TX100 (0.05%), TW20 (0.05%), and a combination of both. The slopes of the blue and green arrows depict how the changes in FI were defined as the initial phase (as shown in **B**) and the late phase (as shown in **C**) of ChAT activity, respectively. Notably, when the initial and the late phase slopes are the same (as for the purple graph in **A**), it means that the activity of ChAT remained unchanged throughout the monitoring time window. (**D**–**F**) illustrate the corresponding data for an identical experiment in the HEPES buffer instead of the TBS buffer. RFU = relative fluorescence unit; TX100 = Triton X100; TW20 = Tween 20; ChAT = choline acetyltransferase; Ctrl = control sample without detergent; BL = blank samples lacking the enzyme, as negative controls.

**Figure 2 ijms-25-13602-f002:**
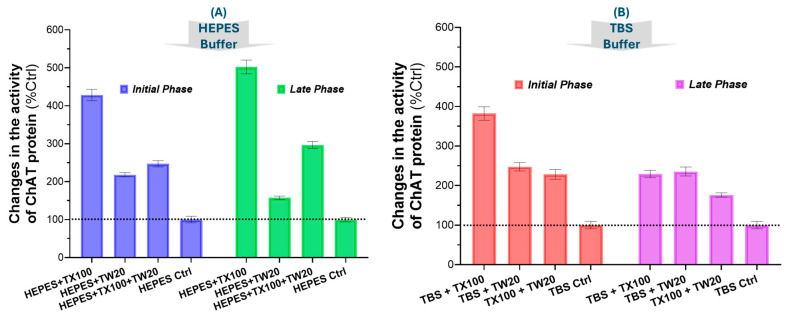
Quantitative changes in the catalytic rate of recombinant human ChAT protein over time in different buffer-detergent conditions. (**A**) The effects of TX100, TW20, or a combination of both on the specific activity of ChAT. The detergents were used at a single constat concentration of 0.05% (*v*/*v*) in HEPES buffer. The buffer without detergent served as control (HEPES Ctrl). The catalytic rate of ChAT was monitored for 30 min, as shown in Figure 1. The data show quantitative changes in ChAT activity within the first 6 min (the initial phase) and the last 20 min (the late phase) of the monitoring time window. (**B**) The quantitative results for a similar experiment setup as in A that was done in TBS instead of HEPES buffer. The data are shown as percentages of ChAT activity in detergent-free control wells (HEPES Ctrl and TBS Ctrl). The bars and the error bars represent the mean and 95% confidence interval (CI) of 14 sample replicates. TX100 = Triton X100; TW20 = Tween 20; ChAT = choline-acetyltransferase.

**Figure 3 ijms-25-13602-f003:**
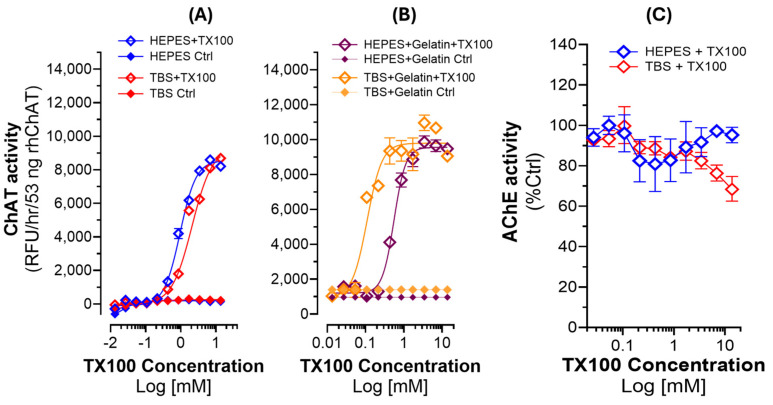
Effect of TX100 on the activity of recombinant ChAT and AChE, assessed across a gradient of TX100 concentrations. (**A**) Canonical S-shaped dose–response curves of the specific enzymatic activity of ChAT with TX100 ranging from 0.8% to 0.008% (*v*/*v*) in the HEPES and TBS buffers. (**B**) The characteristic S-shape dose–response curve from an experiment using the HEPES and TBS buffers that also contained 1.0 mg/mL gelatin, as both a protein stabilizer and a blocking agent. The result also clearly shows that ChAT activity boosted ~10-fold over the detergent-free basal ChAT catalytic rate even when the buffer contained gelatin. Thus, these results confirm that the observed boosting effect of TX100 is unlikely to be due to a mere protein structure stabilization or due to a blocking of ChAT protein adsorption into the surface of the wells of the microtiter plates that were used in the experiments. (**C**) Dose–response analysis of TX100 on recombinant human AChE protein. The result clearly shows that TX100 did not boost the catalytic rate of AChE, as it did for ChAT. TX100 = Triton X100; AChE = acetylcholinesterase; ChAT = choline-acetyltransferase.

**Figure 4 ijms-25-13602-f004:**
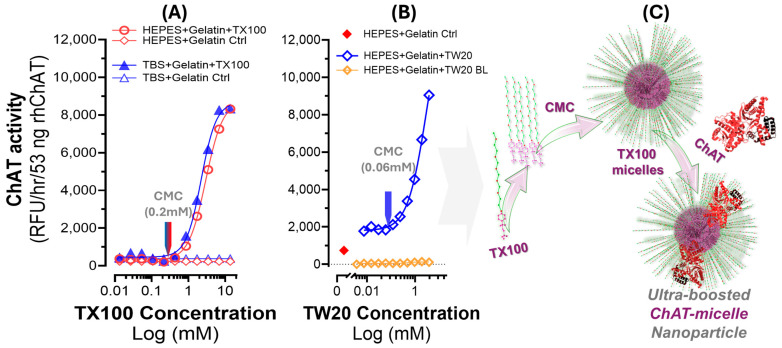
Dose–response curves for TX100 and TW20 versus ChAT catalytic rates. (**A**) ChAT activity in the HEPES– and TBS–gelatin buffer systems in relation to increasing concentrations of TX100. The depicted CMC of TX100 reveals that the boosting effect of TX100 occurs when the TX100 concentration passes the CMC landmark. (**B**) ChAT activity in the HEPES–gelatin buffer systems in relation to concentrations of TW20 around the CMC of this detergent. The CMC of TW20 is depicted in the S-shaped dose–response graph, which reveals that changes in the catalytic rate of ChAT protein by Tween 20 exactly mimic the boosting effect of TX100. In other words, this experiment evinces that, like TX100, the CMC of TW20 is the threshold for the boosting effect of TW20 on the catalytic rate of ChAT. Therefore, CMC appears to be a common denominator for the boosting effect on ChAT protein by these two non-ionic detergents. (**C**) Cartoon depiction of our hypothesis, that as soon as the concentration of the TX100 or TW20 passes the CMC threshold of the detergent, the formation of micelles is propagated through the buffer system, creating membrane-like nanoparticle platforms that mimic the properties of a lipid bilayer cellular membrane, providing a microenvironment that optimally induces an ultrafast catalytic functionality for ChAT protein.

**Figure 5 ijms-25-13602-f005:**
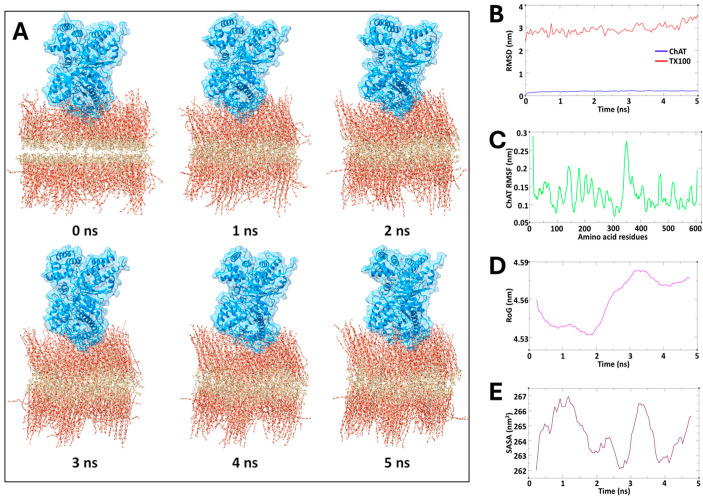
In silico molecular dynamics simulation analysis: (**A**) The spatial configuration of ChAT protein in interaction with a bilayered membrane construct of TX100 micelle along a five-nanosecond MD simulation. Sequential snapshots of the ChAT protein are shown in relation to the TX100 bilayered micelle membrane complex at each nanosecond, from 0 to 5 ns. (**B**) Root mean square deviation (RMSD) plot indicating the overall stability of the protein over time. (**C**) Root mean square fluctuation (RMSF) plot highlighting residue-specific flexibility. (**D**) Radius of gyration (RoG) plot reflecting the compactness of the protein structure. (**E**) Solvent-accessible surface area (SASA) plot illustrating the changes in protein surface exposure to the solvent.

## Data Availability

Data available on request. The data underlying this article will be shared on reasonable request to the corresponding author.

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
