# Peer review of "Micellar Choline-Acetyltransferase Complexes Exhibit Ultra-Boosted Catalytic Rate for Acetylcholine Synthesis—Mechanistic Insights for Development of Acetylcholine-Enhancing Micellar Nanotherapeutics"

_ijms, 2024, doi:10.3390/ijms252413602_

Round 1
Reviewer 1 Report
Comments and Suggestions for Authors
The manuscript ijms-3358638 constitutes a fascinating contribution to the broad domain of molecular biology. Their contribution stands out for the serendipity that it leads to detecting the selective influence of the TX100 buffer on the activity of Choline acetyl-transferase, with a more than notable increase in its activity. Complementary experiences confirm this. Likewise, the authors enrich the discussion by recognizing and referencing analogous contributions.
The Discussion of this work is very carefully prepared and has great clarity. It ends with a paragraph 'In summary,..' (lines 411-415) that makes the scope of the study clear. This paragraph well deserves to be highlighted as 'Point 4. Concluding remarks', which I suggest the authors do without great difficulty.
This work is one more example that in Chemistry and, even more so, Molecular Biology experiments, solvents, buffers, etc. are used under the presumption of innocence. That is, its presence will not have consequences beyond those intended.
Nowadays, we see that such a presumption may be unfounded.
Reviewer 2 Report
Comments and Suggestions for Authors
Choline acetyltransferase is a key cholinergic enzyme responsible for the biosynthesis of acetylcholine, a crucial signaling molecule in non-neuronal cells. Cholinergic dysfunction is linked to neurodegenerative and inflammatory diseases. In the present article, the authors found that Triton X-100 and Tween20 significantly boosted the catalytic rate of human rhChAT protein in buffered solution. Micelle formation is crucial for ultra-boosting ChAT activity, and TX100 or TW20 micelles may facilitate ChAT's full catalytic potential by fine-tuning its structural stabilization and substrate accessibility.
The article is well planned, methods and material are described well, and the results text is corroborated with figures.
Minor suggestions
1. As you mentioned in lines 86-87, TX100 has the potential to alter enzyme functions. Did you encounter this issue during your experiments?
2. Is there any published data demonstrating the potential of TW20 to form membrane-like micellar structures and multilayer organization as you mentioned for TX-100 (lines 326-328)?
3. The discussion could be strengthened by including additional citations of published studies that highlight the potential role of TX-100 and TW20 detergents in micelle formation and the delivery of enzymes.
4. It would be beneficial for the authors to outline the challenges encountered during the use of TX-100 and TW20 in the discussion, as this could provide valuable insights for other researchers.
5. In summary, mention the role of TX-100 and TW20 in drug delivery strategies.
6. In Figures 2 and 3, please ensure that the font size and style for the X and Y-axis legends are consistent.
7. Line 68, Write only the abbreviated form of AD.
8. Make the reference style the same as per journal instructions.
